# Hepatic Involvement of Diffuse Large B-Cell Lymphoma Mimicking Antinuclear Antibody-Negative Autoimmune Hepatitis Diagnosed by Liver Biopsy

**DOI:** 10.3390/medicina59010077

**Published:** 2022-12-29

**Authors:** Euna Lee, Min-Kyu Kang, Gabin Moon, Mi-Jin Gu

**Affiliations:** 1Department of Internal Medicine, Yeungnam University Medical Center, Daegu 42415, Republic of Korea; 2Department of Internal Medicine, College of Medicine, Yeungnam University, Daegu 42415, Republic of Korea; 3Department of Pathology, College of Medicine, Yeungnam University, Daegu 42415, Republic of Korea

**Keywords:** liver cirrhosis, diffuse large B-cell lymphoma, anti-nuclear antibody negative autoimmune hepatitis, liver biopsy

## Abstract

Non-Hodgkin’s lymphoma (NHL) is the fifth most common hematologic disorder in the United States, and its prevalence has been rising in Western countries. Among the subtypes of NHL, diffuse large B-cell lymphoma (DLBCL) mostly involves the lymph nodes, stomach, and gastrointestinal tract, whereas hepatic involvement of DLBCL is rare. On serologic testing, elevated immunoglobulin G (IgG) levels can be observed in DLBCL; however, elevated IgG levels are mainly observed in autoimmune hepatitis. A targeted-lesion biopsy is required for the diagnosis of DLBCL. Based on a final diagnosis, the patient was treated with rituximab-based chemotherapy, including cyclophosphamide, doxorubicin, vincristine, and prednisone chemotherapy (R-CHOP). Herein, we report a case of DLBCL mimicking antinuclear antibody-negative autoimmune hepatitis, which was finally diagnosed as DLBCL involving the liver, and was confirmed by liver biopsy.

## 1. Introduction

The prevalence of hepatic involvement in non-Hodgkin’s lymphoma (NHL) is 15–30% [1,2]. The prevalence of autoimmune hepatitis (AIH) is 4.8 per 100,000 in the population in South Korea, with age adjustment. Antinuclear-antibody (ANA)-negative AIH accounts for 20–30% of AIH cases [3]. Both NHL and AIH have similar laboratory findings, including elevated levels of liver enzymes and immunoglobulin (Ig), including IgG [3,4]. Thus, differentially diagnosing NHL and AIH is difficult. Although liver biopsy is an invasive method, it is a suitable tool for accurate diagnosis when making a definitive diagnosis based on serological findings is difficult [5]. We report a case of NHL mimicking ANA-negative AIH in which liver biopsy played an important role in the diagnosis and treatment.

## 2. Case Presentation

An 85-year-old man presented to our institution with continuous left flank pain, jaundice, and general weakness. He also had symptoms of intermittent nausea and vomiting, with pain worsening on movement. The patient had a history of complete remission of esophageal cancer treated with concomitant chemoradiation therapy. Physical examination revealed no abnormal findings. Contrast computed tomography (CT) revealed liver cirrhosis and marked splenomegaly (approximately 15.5 cm) with a wedge-shaped splenic infarction (Figure 1A).

Neither abdominal lymphadenopathy nor ductal dilatation was not observed. Blood chemistry revealed the following: white-blood-cell count, 6890/μL; hemoglobin level, 7 g/dL; platelet count, 49,000/μL; serum-total-protein level, 8.47 g/dL; serum-albumin level, 1.82 g/dL; total-bilirubin level, 3.62 mg/dL; serum-aspartate-aminotransferase (AST) level, 76 IU/L; alanine-aminotransferase (ALT) level, 22 IU/L; alkaline-phosphatase level, 400 IU/L; gamma-glutamyl transferase level, 59 IU/L; lactate-dehydrogenase(LDH) level, 1665 IU/L; and prothrombin time-international normalized ratio, 1.38. Additional serological tests, including serum hepatitis B surface antigen, hepatitis C antibody, ANA, anti-smooth muscle antibody, antimitochondrial antibody, anti-liver-kidney microsomal type 1 antibody, and other viral tests, including the Epstein–Barr virus, were all negative. However, a peripheral blood smear revealed rouleaux formation of red blood cells. The patient exhibited a markedly elevated serum IgG level of 4889 mg/dL (normal range, 700–1600 mg/dL) and IgM level of 254 mg/dL (normal range, 40–230 mg/dL). Based on radiologic findings, elevated IgG levels, and negative ANA results, we suspected ANA-negative AIH. Additionally, we used the International AIH score, and the results revealed a score of 11, which indicated probable AIH, thus recommending further evaluation. Vibration-controlled transient elastography was performed, and the result was 8.7 kPa, which indicated advanced fibrotic change of the liver without definite findings of liver cirrhosis.

For a clear diagnosis, percutaneous ultrasound-guided liver biopsy was performed (Figure 1B). Histopathological findings revealed small-to-medium-sized atypical lymphoid-cell-infiltration in the portal tract, and the sinusoids were positive for periodic acid–Schiff 5, intermixed with abundant T-cells. CD20 positivity was also observed. To further evaluate atypical lymphoid cells, immunoglobulin heavy-chain-gene rearrangement was performed, and the result was positive, indicating B-cell malignancy. Due to the abundance of T-cells in the biopsy, T-cell receptor gamma-gene-rearrangement was also performed, and the result was negative, which indicated negative T-cell malignancy. Based on the histopathologic results, we excluded T-cell malignancy, rather than B-cell lymphoma (Figure 2A,B)

To evaluate the possible involvement of other organs in lymphoma, positron emission tomography-CT (PET-CT) was performed with a bone-marrow biopsy and the result showed lymphoma involvement of bone-marrow with Bcl-2(+), Ki67(−), MYC(−). The PET-CT revealed F-18 fluorodeoxyglucose uptake in the whole bone-marrow and spleen, and mild uptake in the liver (Figure 3). The final diagnosis was Ann Arbor stage IV diffuse large B-cell lymphoma (DLBCL), with liver, spleen, and bone-marrow involvement (Figure 2C,D).

The patient underwent a reduced-dose regimen of rituximab plus cyclophosphamide, doxorubicin, vincristine, and prednisone chemotherapy (mini R-CHOP), taking into consideration his age. Chemotherapy was considered markedly effective, as his laboratory test-results were as follows: white-blood-cell count, 5360/μL (absolute nuclear count: 4464); hemoglobin level, 9.8 g/dL; platelet count, 72,000/μL; C-reactive protein level, 0.6 mg/dL; serum AST level, 15 IU/L; ALT level, 17 IU/L; and LDH level 338 IU/L (Figure 4). After chemotherapy, he underwent a sufficient recovery period, was in better condition, and underwent regular check-ups.

## 3. Discussion

Based on the initial serological and radiological results, the initial presumptive diagnosis was ANA-negative AIH, with liver cirrhosis. However, for a more precise diagnosis, we performed a liver biopsy, and the result suggested hepatic involvement of DLBCL. Consequently, the patient was treated with systemic chemotherapy. We report a case demonstrating the importance of a precise diagnosis by performing a liver biopsy, which led to appropriate treatment, compared with a presumptive diagnosis.

ANA is the main serological marker for type 1 AIH. However, ANA can be negative in 20–30% of patients [3]. In our case, the patient’s serologic result revealed negative ANA, yet the IgG level was three times higher than the normal range. Recently, IgG4-associated AIH, a novel disease entity that differs from typical AIH, has come into existence, which indicates hepatic accumulation of IgG4-expressing plasma cells (≥10 IgG4 plasma cells per HPF), with markedly elevated serum-IgG levels (≥135 mg/dL) [6]. Although most IgG4-associated AIH had positive ANA or SMA, an autoantibody-negative AIH with clinical symptoms reaching 7% was reported in one study [6]. Therefore, we suspected ANA-negative IgG-associated AIH and used the International Autoimmune Hepatitis Group (IAIHG) scoring system. Our patient scored 11 points in the IAIHG scoring system, indicating probable AIH. Considering the serologic tests and the IAIHG score, we suspected ANA-negative IgG-associated AIH as the etiology of the patient’s chronic liver disease, but not all criteria were typically positive; therefore, we performed a liver biopsy for confirmation. For patients diagnosed with AIH, the treatment is intended to relieve the symptoms and prevent the progression of fibrosis. The first choice of treatment is administering immunosuppression, corticosteroids alone, or a combination of low-dose corticosteroids and azathioprine [7].

Malignant lymphomas are a diverse group of diseases associated with the clonal proliferation of lymphocytes, including Hodgkin’s lymphoma (HL) and NHL [8]. HL accounts for 90% of lymphoma cases, whereas NHL accounts for 10% [9]. NHL is a lymphoid malignancy with variable biological and clinical symptoms, involving the lymphoid and hematopoietic systems [10]. The most common presenting signs and symptoms are abdominal pain and hepatomegaly with fever, weight loss, jaundice, fatigue, and night sweats [11]. DLBCL is the most common type of NHL worldwide, accounting for 30–40% of all NHL cases. DLBCL presents as extranodal disease in approximately 40% of cases, and 10–20% of DLBCL cases involve the bone-marrow [12]. Serological tests generally indicate elevated LDH and beta-2-microglobulin levels [12]. DLBCL can be diagnosed using immunohistochemistry or flow cytometry, and is expressed with B-cell antigens, such as CD19, CD20, and CD22 [12]. When CD20 expression is positive, rituximab is used for treatment. DLBCL is known as chemotherapy-sensitive cancer, and the most commonly used regimen is R-CHOP and alternating triple-combination chemotherapy (ATT), which includes an alternating combination of (1) methotrexate, bleomycin, doxorubicin, cyclophosphamide, vincristine, and methylprednisolone and (2) mesna, ifosfamide, mitoxantrone, and etoposide with (3) doxorubicin, methylprednisolone, cytarabine, and cisplatin can be also used. If it is a primary extranodal-involved DLBCL, chemotherapy or a combination of surgery with radiotherapy (RT) can be considered as an option. In recent studies, hypofractionated RT and low-dose RT have been used to treat recurrent tumors of DLBCL in need of palliation [13]. At the molecular level, it is known that the specific gene expressions can predict the prognosis of DLBCL; for example, the expression of both MYC and BCL2 (“dual-expresser”) by immunohistochemical staining has been associated with poor prognosis [14]. The presentation of lymphoma involving other organs, especially the liver, may vary with the incidental discovery of hepatic abnormalities and from asymptomatic patients to fulminant hepatic failure, with the rapid progression of encephalopathy [1]. To diagnose DLBCL involving the organs, PET-CT was formally incorporated into the standard staging system [15]. These cross-sectional imaging modalities have superseded staging laparotomy or splenectomy, by demonstrating abdominal nodal groups and organ involvement [16]. Our patient also underwent PET-CT; the results revealed diffuse mild uptake in the liver, and liver biopsy was recommended for confirmation.

Liver biopsy is considered the gold standard for diagnosing cirrhosis and the staging of liver fibrosis, as well as for other liver diseases. Despite advances in noninvasive tests, liver biopsy remains an important method for diagnosing and evaluating diffuse liver disease among patients with hepatopathy of unknown origin, and assessing the severity and chronicity of cirrhosis by quantifying fibrosis and architectural changes [5,17]. Several previous studies have demonstrated that liver biopsy plays a crucial role in changing the diagnosis and treatment of patients with ambiguous liver diseases. In a previous large cohort study that enrolled 2413 patients, the study revealed that 31.2% of the diagnoses were changed by confirmation through liver biopsy [18]. In a retrospective study of liver biopsy performed on 259 patients with chronic liver disease, the liver biopsy changed the diagnosis and helped in performing clinical management in 70% of the patients [19]. There was even a case with unexplainable abnormal liver function with no abnormality in the image study but which was finally diagnosed as a primary liver lymphoma after performing a liver biopsy [20]. In terms of the safety of liver biopsy, major complications of liver biopsy, such as bleeding (causing hypotension and hematoma, requiring blood transfusion), can occur in 1–2% of the patients, and post-procedure mortality is reportedly <0.001–0.2% [21]. Although biopsy can be an invasive method, the incidence of related complications is very low.

A limitation of this study was the lack of laboratory tests. When patients have elevated IgG levels, further evaluations, such as IgG subclass, should be performed. If the LDH level is increased, suspected lymphoid disease and beta-2-microglobulin can lead to an earlier diagnosis.

## 4. Conclusions

In this case, initially, we suspected ANA-negative AIH, based on the laboratory findings (elevation of IgG, bilirubin, AST/ALT) and the image study. Treatment of AIH is mainly by steroid or an immunosuppressive agent such as azathioprine. In DLBCL, the treatment of choice is chemotherapy (R-CHOP). It can be difficult to diagnosis differentially these two diseases just by laboratory findings or image studies. However, it is important to make an accurate diagnosis, because the direction of treatment varies, depending on the diagnosis. Furthermore, as in this case, the diagnosis could turn into an even more aggressive form, including cancer. Therefore, we report this case to emphasize the importance of liver biopsy as a diagnostic tool for ambiguous presentations of liver disease; only histologic confirmation of the target lesion can lead to a precise diagnosis and proper treatment.

## Figures and Tables

**Figure 1 medicina-59-00077-f001:**
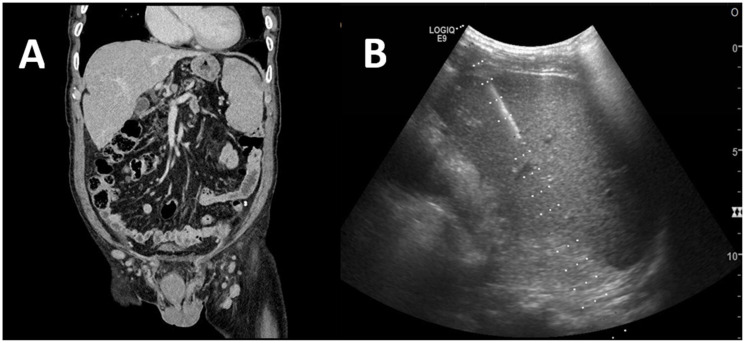
Imaging study on admission. (**A**) Contrast-enhanced computed tomography revealed hepatomegaly with cirrhosis-like surface of liver and splenomegaly with peripheral, wedge-shaped hypo-enhancing region, which indicates typical splenic infarction. (**B**) Percutaneous ultrasound-guided liver biopsy was performed.

**Figure 2 medicina-59-00077-f002:**
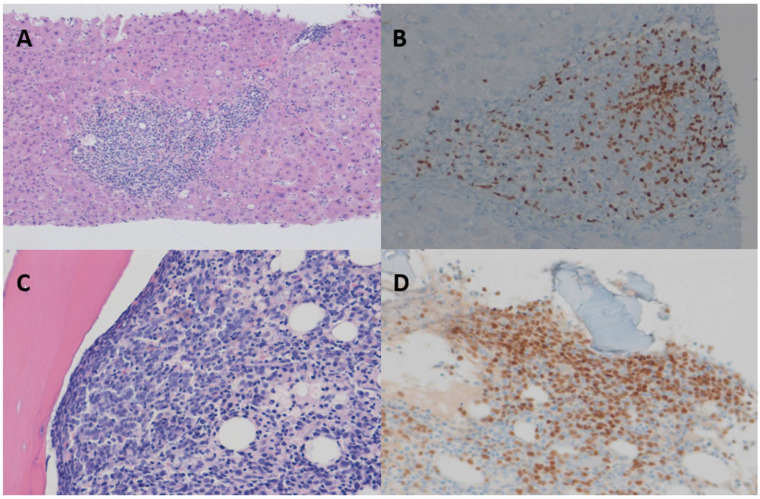
Histologic findings of the liver. (**A**) Liver biopsy revealed atypical lymphocyte infiltration in the portal tract (hematoxylin–eosin [HE] staining, ×10, respectively). (**B**) Medium-sized atypical lymphocyte infiltration with positive periodic acid–Schiff (PAS) 5 (×100). (**C**) Bone-marrow biopsy revealed large atypical lymphoid infiltrating bone-marrow (HE staining, ×200, respectively). (**D**) Medium-sized atypical lymphocyte infiltration with positive PAS 5 (×200).

**Figure 3 medicina-59-00077-f003:**
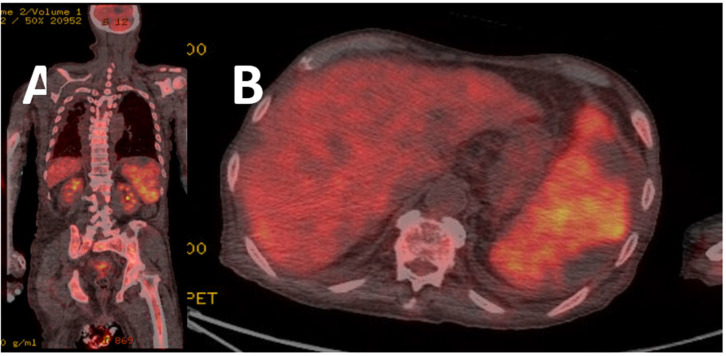
Results of the positron emission tomography–computed tomography (PET–CT). (**A**) Coronal view of the PET–CT, indicating diffuse F-18 fluorodeoxyglucose (FDG)-uptake in bone-marrow. (**B**) Axial view of the PET–CT, revealing splenomegaly with diffuse FDG uptake, wedge pattern of photon defects in low density indicating lymphoma involvement concomitant splenic-infarction with diffuse- and mild-FDG-uptake in the liver.

**Figure 4 medicina-59-00077-f004:**
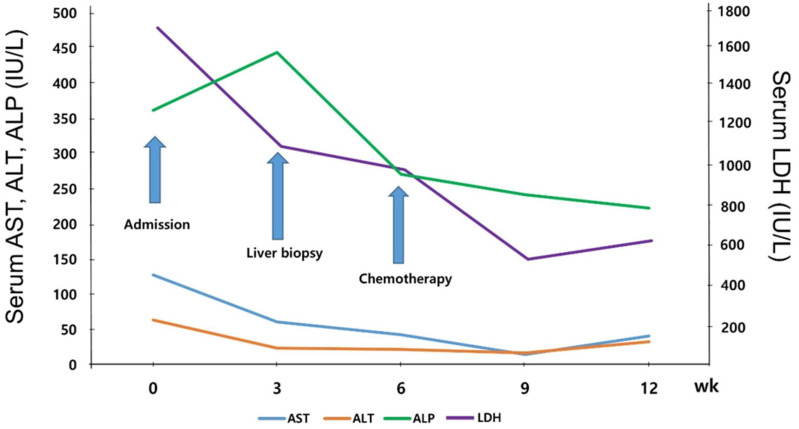
Clinical course of the patient. Aspartate aminotransferase (AST), alanine aminotransferase (ALT), alkaline phosphatase (ALP), and lactate dehydrogenase (LDH) levels markedly decreased after chemotherapy.

## Data Availability

The data supporting the findings of this study are also available from the corresponding author (M.K.) upon reasonable request.

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
