# Peer review of "Hepatic Involvement of Diffuse Large B-Cell Lymphoma Mimicking Antinuclear Antibody-Negative Autoimmune Hepatitis Diagnosed by Liver Biopsy"

_medicina, 2022, doi:10.3390/medicina59010077_

Round 1

Reviewer 1 Report

The authors reported an interesting and a well-presented case of diffuse B-cell lymphoma with hepatic involvement. However, some issues should be addressed:

1) The authors should highlight the novelty of the case reported and some particularities, in the light of published data on DLBCL with hepatic involvement (e.g., https://www.ncbi.nlm.nih.gov/pmc/articles/PMC5922499/, https://doi.org/10.1016/j.jcrpr.2017.07.004, https://doi.org/10.1155/2016/2053257, https://www.ncbi.nlm.nih.gov/pmc/articles/PMC3206687/), including a recent review (https://www.sciencedirect.com/science/article/pii/S2772736X22000652);

2) Some details regarding management of the case (including evolution of the patient) should be reported in the abstract (e.g., chemotherapy). Also, a conclusion should be mentioned in the abstract, to raise a issue for future research or to summarize the main findings;

3) Conclusions should more detailed, to summarize the main findings, some particularities of the case, issues on diagnosis or management, rather than to highlight only the importance of liver biopsy;

4) Diagnostic and therapeutic management workup should be discussed comparatively with other published studies and case-reports (including those mentioned above), to provide an overall direction for future research and guidelines (in the discussion section);

5) Different therapeutic approaches, like chemotherapy (including different drug regimen reported in other clinical studies and guidelines) or surgery should be discussed;

6) The patient’s prognosis should be also discussed, including some clinical/biological/genetic (including some mutations) features linked to worse or better response to therapy and evolution.

Author Response

The authors reported an interesting and a well-presented case of diffuse B-cell lymphoma with hepatic involvement. However, some issues should be addressed:

1) The authors should highlight the novelty of the case reported and some particularities, in the light of published data on DLBCL with hepatic involvement (e.g., https://www.ncbi.nlm.nih.gov/pmc/articles/PMC5922499/, https://doi.org/10.1016/j.jcrpr.2017.07.004, https://doi.org/10.1155/2016/2053257, https://www.ncbi.nlm.nih.gov/pmc/articles/PMC3206687/), including a recent review (https://www.sciencedirect.com/science/article/pii/S2772736X22000652);

We really appreciate the caring review of our manuscript. We appreciate the time and effort that you have dedicated to providing valuable comments on our manuscript. We went through the reviewer attached files and we added more contents about our particularities in discussion and conclusion. (Page 6, line 197~207)

Primary hepatic lymphoma (PHL) defined as lymphoma confined to the liver without any involvement of the spleen, lymph nodes, bone marrow, peripheral blood, or other tissues. In our case, patient has been revealed involvement of lymphoma in spleen and bone marrow, so he was suitable for secondary lymphoma. But we founded very helpful information about DLBCL involving liver through reviewer’s attached files and added citation. Thank you for your review.  

2) Some details regarding management of the case (including evolution of the patient) should be reported in the abstract (e.g., chemotherapy). Also, a conclusion should be mentioned in the abstract, to raise a issue for future research or to summarize the main findings;

I totally agree with the reviewer's opinion. We have added the text at abstract about the treatment of the patient and the importance of the liver biopsy. Again, thank you for your careful review.   (Page 1, line 18~19)  

3) Conclusions should more detailed, to summarize the main findings, some particularities of the case, issues on diagnosis or management, rather than to highlight only the importance of liver biopsy;

Thanks for your detail comments. We described more in detail on conclusion (Page 6, line 197~207) about the process of our case and reasons we performed liver biopsy at the end.

4) Diagnostic and therapeutic management workup should be discussed comparatively with other published studies and case-reports (including those mentioned above), to provide an overall direction for future research and guidelines (in the discussion section);

 Thank you for your valuable comment and the attached files. Through attached files, there was a similarity recommending early liver biopsy when patient’s laboratory findings or image studies are abnormal but when etiology is uncertain. We emphasized the importance of liver biopsy for accurate diagnosis again and we hope more institutions also consider liver biopsy more aggressively. (Page 6, line 185~187) Thank you for your comment.

5) Different therapeutic approaches, like chemotherapy (including different drug regimen reported in other clinical studies and guidelines) or surgery should be discussed;

Thank you for your caring comments. We have searched more journals and papers about other treatable regimen, such as alternating triple-combination chemotherapy (ATT) and other up to dated treatment such as radiotherapy and added in Discussion section (page5, line 154~162). Unfortunately, our case was about secondary liver involvement of lymphoma rather than primary, so surgery was not a considerable option. Thank you again for your comment.

6) The patient’s prognosis should be also discussed, including some clinical/biological/genetic (including some mutations) features linked to worse or better response to therapy and evolution

Thank you for your comment. We searched more information about DLBCL prognosis in molecular level which can predict prognosis of DLBCL, and we inserted more citation in discussion. (Page 5, line 162~165) Thank you for your opinion.

Thank you again for your kind and valuable comments. We hope that our revision will meet with approval. We would like to respond to any further questions and comments you may have.

Reviewer 2 Report

Dear authors, I read the manuscript and have the following comments:

- detail the case particularity and the clinical rationale that lead to the initial diagnostic of  ANA-31 negative AIH

- please provide more clinical data on patient's symptoms and prior investigations

- What was the initial rationale and how did you suspect the diagnostic of AIH?

- please explain the largely pallet of investigations (PET-CT, VCTE, extended autoimmune panel) in contrast with not performing liver biopsy in the first place.

Author Response

Dear authors, I read the manuscript and have the following comments:

1) detail the case particularity and the clinical rationale that lead to the initial diagnostic of  ANA- negative AIH

Thank you for your caring comments. We added more detailed information about the patient’s clinical symptoms and citation the rationale about ANA negative IgG associated AIH. Our patient showed markedly increased IgG level with hyperbilirubinemia and AST/ALT elevation but negative in another autoimmune serologic lab. ANA negative IgG AIH is emerging term recently not less in prevalence, so we suspected this as a first diagnosis and performed liver biopsy for confirmation. (Page 5, line 127~133) Thank you again for your comment.

2) please provide more clinical data on patient's symptoms and prior investigations

 We totally agree with the reviewer’s opinion. We added more clinical symptoms of patient in case presentation. Thank you for your comment. (Page 1, line 37~39)

3) What was the initial rationale and how did you suspect the diagnostic of AIH?

4) please explain the largely pallet of investigations (PET-CT, VCTE, extended autoimmune panel) in contrast with not performing liver biopsy in the first place.

Thank you for your comment. When the patient first visits our hospital, he’s main symptom was left flank pain and jaundice, so we examined abdominal CT and CT finding showed hepatomegaly and splenic infarction and he’s lab finding showed hyperbilirubinemia with AST/ALT, LDH elevation. To find a cause of hepatomegaly and lab finding, we suspected liver cirrhosis, so we performed VCTE. But etiology of cirrhosis (even though the patient revealed not having cirrhosis at the end) was not clear. So, we decided to do autoimmune panel to find an etiology of cirrhosis and found out that IgG level was extremely high. There was a western study, based on Umemura’s histologic criterion, the prevalence of IgG4-associated AIH was 25%(https://www.sciencedirect.com/science/article/pii/S1590865815006519?via%3Dihub). Taken together, there was no findings of obstructive jaundice, no sign of cirrhosis but IgG level, bilirubin, AST/ALT was still elevated so diagnosis was ambiguous. To clarify the diagnosis, we decided to perform liver biopsy and it turned out to be as lymphoma involving liver. We had to evaluate other organ or bones of lymphoma involvement, so we performed PET/CT and bone marrow biopsy and result showed he had liver, spleen and bone marrow involved by lymphoma.

We did not perform liver biopsy as a first line modality because even though liver biopsy is a minor complication procedure but still has a chance of complication so we performed other non-invasive evaluations in advance.

Thank you again for your kind and valuable comments. We hope that our revision will meet with approval. We would like to respond to any further questions and comments you may have.

Round 2

Reviewer 2 Report

thank you for the additional data, I agree with the present form